# Wideband Multiport Antennas

**DOI:** 10.3390/s20236960

**Published:** 2020-12-05

**Authors:** Mehdi Seyyedesfahlan, Abdulkadir Uzun, Anja K. Skrivervik, Ibrahim Tekin

**Affiliations:** 1Microwaves and Antennas Group (MAG), EPFL, CH-1015 Lausanne, Switzerland; anja.skrivervik@epfl.ch; 2Electronics Engineering, Sabanci University, 34956 Istanbul, Turkey; kadiruzun@sabanciuniv.edu; 3ASELSAN A.Ş., 34906 Istanbul, Turkey; 4Sabanci University Nanotechnology and Application Center (SUNUM), 34956 Istanbul, Turkey

**Keywords:** wideband antenna, MIMO antenna, four-port wideband antenna

## Abstract

In this paper, a wideband four port 2–6 GHz antenna is proposed. One-, two-, and four-port antennas are implemented and characterized between 2 and 6 GHz. The isolation between the ports is improved by connecting and optimizing the ground plane sections. The results show that the antennas’ reflection coefficients are better than 10 dB in the frequency band. The measured isolation between the ports is greater than 15 dB (between 2.3 and 6 GHz) and 10 dB in the whole band for two- and four-port antennas, respectively, however, it is more than 20 dB around 2.4 and 5–6 GHz for both antennas. The calculated correlation coefficient between ports is below −30 dB (>2.14 GHz) and −15 dB for the two- and four-port antennas, respectively. The measured gain and efficiency scale are 3.1–6.75 dBi and 62–98%, respectively. To the best of our knowledge, an antenna both being wideband from 2 to 6 GHz and having independent four ports is only addressed in this work. The four-port antenna can be used for MIMO systems or smartphones operating on many wireless systems simultaneously such as 3G/4G/5G Sub-6 GHz and WLAN including the next generation WiFi7 with full-duplex operation.

## 1. Introduction

Recently, there is an increasing demand for higher throughput and more reliable transceiver systems with an application on 4G wireless systems and mobile communication. Multiple Input Multiple Output (MIMO) technology could be a promising candidate for this purpose [1]. The MIMO technique is based on using multiple antennas to increase the data rate by means of uncorrelated signals.

For the MIMO system to function as expected, the mutual coupling between antenna elements should be as low as possible. A standard approach to achieve MIMO operation is to develop multiple antennas that are sufficiently separated to achieve the desired level of signal independence and port-to-port isolation. However, this will make the transceiver system bulky and result in increased assembly costs. Additionally, ease of integration and miniaturization are two major challenges ahead of MIMO antennas. Thus, the design of the MIMO antenna is the first important thing to be addressed to improve the overall system performance. Planar type antennas are preferred for MIMO applications due to ease of integration and low cost. For miniaturization purposes, there are no options but to space antenna elements closer or designing multiport, single-element antenna. Various studies have been carried out aiming to design such compact antenna systems [2,3]; they are commonly based on the planar antenna prototype [4,5]. The first approach is to decrease the spacing between antennas and keep the mutual coupling at an acceptable level by applying isolation improvement techniques. Examples of this approach can be found in various literature, such as adding a ground wall with connecting line and shorting pins [6], T-shaped ground plane [7], the corrugated ground plane with λ/4 slot [8], modified PIFA with a small local ground plane [9], techniques based on dispersion engineering called negative group delay (NGD) technique [10], use of external lumped element decoupling networks between the feed ports to allow matching of even and odd modes to a common impedance and thereby producing small cross-correlation and maximum gain over a limited frequency range [11], and other compact designs of MIMO antennas [2,3,4,5,12]. These methods can reduce the overall size of antennas and insulator regardless of the difficulty in insulator design.

The second approach takes advantage of multiport, single-element antennas to propose a more compact solution. A novel design of dual-feed, single-element antennas for 4G MIMO terminals is proposed and analyzed in [13]. The antenna consists of a radiating patch which is fed by two input ports. The idea is to use an isolated mode antenna (iMAT) [14] to reduce the antenna size and mutually couple the ports. The iMAT works based on exciting and different propagating modes of antenna for different ports. The iMAT antenna idea is also used in [15] to design a novel u-shaped single-element antenna with better performance, compared with two separate monopole antennas, in [16,17] to design a multiple compact multimode patch antenna, and in [18,19,20] for a multimode antenna that is not based on a patch antenna. 

An important factor in MIMO systems is its bandwidth, which is determined by the bandwidth of the antenna element. Thus, a wideband single-element antenna with multiple ports could be very useful for a wideband MIMO system. In general, MIMO systems use many antennas to obtain multiport systems. Note that a multiport wideband antenna can also be used in smartphones that use many different wireless protocols at different frequency bands at the same time. There exist antennas which either are wideband or are multiport with narrow bandwidth. Nevertheless, combining multiport with wide bandwidth operation forms our antenna’s novelty, which has four ports and can operate between 2 and 6 GHz. We propose a structure to increase the number of radiating element feeding/receiving ports only by rotating the main single port monopole antenna. Of course, monopole antenna is well known and there are many reports on how to make it wideband; however, increasing the number of ports while matching the ports and decreasing coupling between the ports requires many attempts. Moreover, when all the ports use common radiating elements, it needs a smart method to mitigate coupling between the ports. In this paper, we use a unit structure and bridge between the ground planes of ports to alleviate coupling between ports. We designed and optimized the antenna for frequency band between 2 and 6 GHz and achieved minimum isolation of 10 dB between the four ports. The aim is to introduce a multi-purpose (multiport and wideband) structure; however, for the desired application/band, the isolation between ports can be increased only by optimizing the ground plane and connection between ports.

The four-port antenna reported in this paper can be used for a multi-frequency system requiring many antennas. The 4 × 4 MIMO implemented for a WLAN on 2.4 and 5.2 GHz band is one example. The four-port antenna can also be used for the sub-6 GHz band 5G system. For a multiple radio system currently used in smartphones, let us assume there are four radios and these radios are 3G (2 GHz band), WLAN (2.4 GHz), 1–6 GHz 5G (3.6 GHz band), and WLAN (5.2 GHz). One can directly connect these four radios to the proposed four-port antenna without any switches and duplexers. RF filters can be deployed for each radio band to provide enough selectivity. However, with our antenna, all these radios can operate simultaneously. The key focus is on new mobile 5G bands including spectrum in the 3.5 GHz range that has been assigned in numerous countries. However, several countries including China and Japan plan to use spectrum in the 4.4–4.9 GHz range for 5G in addition to a growing number of countries considering the 3.5–4.2 GHz range, as well as the 2.3 and 2.5/2.6 GHz bands for 5G NR [21].

To have a wideband multiport antenna, a wideband planar structure should be selected. In this work, a printed monopole disk antenna [22] is selected for multiport use, due to its wide bandwidth operation. Figure 1 shows the monopole disk antenna with a single port. The disk monopole antenna is modified to two- and four-port versions for different frequency ranges. The geometrical symmetry of the antenna shape not only makes the design easy but also gives the versatility of adding and increasing the number of ports. 

In this paper, two metrics are used for the assessment of the isolation between antenna ports: the S parameter and the correlation coefficient. The correlation coefficient expresses antenna pattern independence to the S parameter, which is necessary for a MIMO antenna. This paper is organized as follows. Section 2 demonstrates the design of single-, dual-, and quad-feed disk monopole antenna with wideband operations. Section 3 presents and compares the simulated and measured results for the S parameter as well as the radiation patterns of the antennas. Section 4 summarizes and concludes the paper.

## 2. Multiport Antennas Design

In this section, the design process for the one-, two-, and four-port antennas are introduced. The antennas contain a radiating disk and microstrip transmission line as the antenna feed. Both two- and four-port antennas have structures similar to the single-port antenna, and the various dimensions shown in Figure 1 are optimized for each antenna to match each port to 50 Ω and decrease the mutual coupling between ports of each antenna, over the frequency band of 2–6 GHz. The scheme for increasing the number of the ports is to exploit the single-port antenna geometry (Figure 1) as the basis of n-port antennas, and then rotate/add the structure by 90° (with respect to disk center) to form the new port. The advantage of this procedure is that the ports (in multiport types) would be similar, and the design parameters in Figure 1 are optimized for all ports, simultaneously. The optimization is performed to approach the specified reflection coefficient and isolation between the ports over the desired frequency bandwidth. The antennas were simulated and prepared for fabrication on d = 0.787 mm thick (copper cladding tc = 35 µm) Rogers RT/duroid 5880 laminate with a dielectric constant of 2.2 and tangent loss of 0.0009.

### 2.1. Single-Port Antenna

The schematic of the single-port antenna and the parameters for which optimizations are performed are shown in Figure 1. The antenna can be divided into two major parts: the radiating disk and the transmission line that feeds the disk. The fabricated antenna with the dimensions of 6.8 cm × 4.4 cm is shown in Figure 2. In the bottom layer, an incomplete triangular shape ground plane supports the signal line in the top layer and can have coupling with the radiating disk.

The dimension of the disk (*r*) adjusts the antenna operating frequency, while Δ*r* is the spacing between the disk (top layer) and ground plane (bottom layer) edge. Since the antenna is similar to a monopole antenna, the disk will resonate with a quarter-wavelength diameter (2*r* = λ/4). The radius of the disk for resonating at 2 GHz in the free space is calculated as 18.75 mm, which is used as the initial value for r. To match the antenna to 50 Ω in the band of 2–6 GHz, other parameters (shown in Figure 1) are utilized to tune the antenna over the entire desired frequency band or in some specific frequencies. *θ* and *h_p_* control the dimensions of the ground plane. The gap specified by the dimensions of *g_a_*/*g_b_*, as well as the location (*h_d_*) and dimensions of the dumbbell-shaped etching, affect the antenna reflection coefficient by changing the inductance/capacitance of the transmission line and improving the feed line S_11_ magnitude.

The optimized parameters for fabricating the antenna (Figure 2) are reported in Table 1. Different parameters of the single-port antenna are swept around the optimized values to show their effect on the antenna reflection coefficient.

Figure 3a shows that the optimum value for a disk radius of 21 or 22 mm can give the best reflection coefficient values at less than −15 dB. As the spacing between the disk and ground plane is increased up to 3 mm, the antenna matching is improved, while greater values deteriorate the antenna performance, due to the decoupling between the microstrip line and the disk antenna, as shown in Figure 3b.

When the ground plane angle (*θ*) is increased, the antenna |S_11_| is improved for higher frequencies, while the impedance matching worsens in the middle of the band (Figure 4a). As shown in Figure 4b, the height variation of the truncated triangle ground causes a frequency shift in the antenna reflection coefficient.

The effect of the dumbbell-shaped etched ground plane on matching the antenna between 2 and 6 GHz is demonstrated in Figure 5a. Note that the legend with the word “No” in Figure 5a points to the full ground (without dumbbell-shaped etching). Changing the dimensions of the etched area in Figure 5b,c shows its major effect on the antenna reflection coefficient for frequencies greater than 3 GHz. Although the effect of some parameters is not that significant in the single-port antenna, they play a drastic role in tuning the multiport antennas in the wideband operation.

### 2.2. Two-Port Antenna

The two-port antenna is obtained by rotating the single-port antenna by 90°, with respect to the center of the disk, and adding another port. As shown in Figure 6, the ground planes of the two ports are connected via a circular ring sector. 

The angle of the sector is 90°−*θ*, while its inner and outer radius are *R_d_* and *h_p_*, respectively. By connecting the grounds of the two ports, better isolation between the ports is obtained. When the first port of the antenna is fed, the received signal in the second port includes two components: (a) the signal that passes over the disk; and (b) the signal that flows from the connected ground of the ports. Therefore, these two components can cancel each other, if the phase difference of 180° is kept when these two components arrive at the second port. Out of phase condition between the mentioned two current trajectories improves the ports’ isolation significantly and can be achieved by optimizing some of the antenna parameters. The working mechanism of the connection is shown (see Figure 7) in the simulated current distribution on the antenna at 2.4 GHz and at different phases.

Note that the power that is dissipated in the vicinity of port two (due to cancellation) can decrease the radiation efficiency of the antenna, whereas the used PCB board is chosen to have a very small tangent loss. The dimensions of the fabricated two-port antenna are 7.6 cm × 7.6 cm and the rest of the parameters are given in Table 1. 

Changing the disk radius (*r*) and Δ*r* can both affect the accepted/reflected power by the first port on the disk side as well as the coupled power to the second port through the disk. The influence of this complicated process on the insertion loss between the ports, for various *r* and Δ*r*, is shown in Figure 8.

As shown in Figure 8, the most significant effect of the parameters *r* and Δ*r* on |S_21_| is between 2.7 and 4.5 GHz. At these frequencies, insertion loss can be adjusted to be below −20 dB, by the disk size and the gap spacing. The ground plane angle also affects insertion loss between ports in a limited frequency band of 4–5 GHz (Figure 9a). Increasing the height of the ground plane shifts the |S_21_| to lower frequencies in the 3–6 GHz band (Figure 9b).

As discussed above, the connection between the ports’ ground plays an important role in improving the insertion loss between them. The important factor *R_d_*, which controls the dimension of the connected part, and corresponding isolation between the ports, is swept around its optimum value *R_d_* = 36 mm, as shown in Figure 10. This results in increasing *R_d_*, and hence decreases the thickness of the connected section and causes the S_21_ curve to shift to lower frequencies.

### 2.3. Four-Port Antenna

Another multiport antenna is a four-port antenna that is formed by rotating/adding a single-port antenna with respect to the disk center. In this type of antenna, the spacing between the ports is 90°, as shown in Figure 11.

For this antenna, the basic geometry is similar to that of the single port, while the edge of the disk at the ports is etched (in top view) and the grounds are connected using a metal bar with a thickness of *C_t_*. The etched areas on the disk are a rectangular section with dimensions of *e_a_* and *e_b_* (Figure 11), which alleviate the ports’ reflection coefficient to be below −10 dB within the band of 2–6 GHz. The ground connection also controls the insertion loss between the ports, the same technique as used in the two-port antenna. The dimensions of the fabricated four-port antenna are 12.4 cm × 12.4 cm, with parameters given in Table 1. Note that, for an application on a mobile phone, antenna size can be made smaller by bending from the microstrip line sections. The proposed four-port antenna is designed for a wide frequency band, starting from 2 GHz. By excluding WiFi 2.4 GHz frequency, while shifting start frequency to 3 GHz, which means, if only 5G systems are chosen, the disk size and hence the overall antenna size will be smaller by a factor of 1.5 times to achieve an 8 cm × 8 cm antenna. Moreover, for mobile applications, part of the feeding network can also be placed in a different PCB layer, or a flexible board may be folded/wrapped and antenna size can be further made smaller. Moreover, the antenna can be optimized/improved by separating 5G or WiFi system and having two antennas. For example, for a 5G sub-6 GHz system, the antenna size can be optimized to 3 GHz, and for WiFi it could be around 5 GHz band.

As shown in Figure 12, etched areas on the disk (geometrical parameters *e_a_* and *e_b_* in Figure 11) and the ground plane (geometrical parameters *g_a_* and *g_b_* in Figure 1) play a very crucial role in adjusting each port’s reflection coefficient below −10 dB. The legend entry “No” indicates no etching is performed on the copper.

Due to the wide bandwidth of 2–6 GHz and the number of ports, developing the antenna for |S_11_| < −10 dB and desired isolation between the port is challenging or maybe impossible for some geometries. Consequently, it is proposed to match the ports to 50 Ω with |S_11_| < −10 dB and improve ports isolation for some specific frequencies, while it exceeds 10 dB for whole the bandwidth. Therefore, isolation between the ports is optimized to target the higher values around the frequency of 2.4 GHz and bandwidth of 5–6 GHz that are used by WLAN.

Since the ports are symmetric, S_21_ = S_41_ = S_32_ = S_43_, and S_31_ = S_42_, only S_21_, and S_31_ is plotted. As shown in Figure 13, as the ground plane angle (*θ*) is increased, the magnitudes of the S_21_ and S_31_ shift to higher frequencies.

The increasing *θ* causes an increase in the ground plane size, and, as a result, the length of the connected part between ground planes of the ports is decreased. Moreover, increasing the height of the ground plane (*h_p_*) increases the length of the connected part and shifts S_21_ and S_31_ to lower frequencies (see Figure 14).

When the thickness of the connected part (*C_t_*) is increased, as in Figure 11b, since the lower side of the connection part is limited/fixed by the triangular-shaped plane (*h_p_*), the upper edge is extended toward the disk. Therefore, by increasing the thickness of the connected part (*C_t_*), its average length is decreased, shifting S_21_ and S_31_ to higher frequencies (see Figure 15).

## 3. Measurements and Simulations

The antennas were simulated, fabricated, and measured using the geometrical parameters in Table 1. The antennas were characterized for S parameters and 3D cross-polar and co-polar gain at some specific frequencies. The correlation coefficient between ports i and j of N-port antennas is calculated using the simulated and measured S parameters using (1). Equation (1) is an approximation to calculate the pattern independence between the ports using the S parameter. Its precision is increased as the radiation efficiency of the antenna is increased [23]
(1)$$\rho_{e}(i,j,N)=\frac{{\left|C(i,j,N)\right|}^{2}}{\prod_{k=i,j}\left[1-C(i,j,N)\right]},\;\;C(i,j,N)=\sum_{n=1}^{N}S_{i,n}^{*}S_{n,k}$$

The measurements were performed at Sabanci University Anechoic Chamber that is suitable for the frequency range from 700 MHz to 50 GHz and is equipped with a PNA5245A vector network analyzer (working up to 50 GHz).

### 3.1. Reflection Coefficient

Simulated and measured reflection coefficient results from 1 to 6 GHz of the single-port antenna are shown in Figure 16. The measured |S_11_| has some shifts for frequencies greater than 4 GHz. This shift can result from the PCB dielectric constant variation in different frequencies or the effect of the measurement setup. Although the antennas are measured inside the anechoic chamber, due to their isotropic radiation pattern (which is discussed in the next section), the absorbers, feeding cable, and setup in close distance to the antenna can affect its performance. The measurement shows that the antenna reflection coefficient is below −10 dB for the whole 1–6 GHz frequency band.

The measured and simulated S parameters and calculated correlation coefficient using (1) for the two-port antenna are shown in Figure 17. Some frequency shift around 200 MHz is also seen in the measured S parameters. The measured results comply with the simulated ones and the antenna is matched to 50 Ω for frequencies greater than 1.1 GHz. Isolation between the ports is better than 15 dB for higher frequencies (>2.3 GHz). After a frequency of 2.14 GHz, a correlation coefficient of better than −30 dB is obtained from the measured/calculated ρ_21_.

The four-port antenna is characterized for S_11_, S_21_, and S_31_, as shown in Figure 18. The measured and simulated results agree well, and only some frequency shift is seen in S_21_. As mentioned for the two-port antenna, the frequency shifts between simulation and measurement can be the effect of setup (such as cables) that reflect back the radiated field from the antenna and change the antenna performance. When cables are used for measuring two close ports such as S_21_, their influence is more pronounced than for the other ports such as S_31_. In addition, as the radiating disk is surrounded by the metal ground plane (the number of ports is increased), the antenna radiation on the ground plane direction is reduced. Thus, the effect of any cables, which are extended in the same plane as the ground plane, is decreased by increasing the number of ports. 

The measurements show that the S parameters are below −10 dB over the desired frequency band of 2–6 GHz. Isolation between the ports is better than 20 dB at around 2.4 GHz and between 5 and 6 GHz. In addition to the S parameters, good agreement between the measured and simulated correlation coefficient is also obtained (see Figure 18b).

For MIMO applications, although there are no specific requirements on isolation values, lower values of isolation will ease the work done by the baseband processor. Two- and four-port antennas can operate at frequencies below 6 GHz with good isolation values. For the two-port antenna, isolation is lower than 15 dB in the overall band mainly around 20 dB, which may be sufficient for MIMO applications. For the four-port antenna, isolation is lower than 10 dB in the overall band, more than 20 dB around 2.4 and 5–6 GHz, and more than 10 dB in the whole band. Further, for a 5G MIMO system, the four-port antenna can also be used for some portions of the bandwidth. The least isolation is 10 dB; however, the 5G sub-6 GHz system will not use the whole 4 GHz available, but a few hundred MHz bandwidth from the spectrum. When we consider a realizable 5G massive MIMO system with a few hundred MHz bandwidth operations, the four-port antenna may achieve more than 20 dB isolation, which may be sufficient for a MIMO system such as in 5–6 GHz band.

There is also interest in how these antennas will perform in a real environment. Most of the time, measured results are performed in a controlled environment such as an anechoic chamber. When these antennas are placed in a real environment, the isolation, as well as the return loss of the antenna, may change. However, most of these isolation and reflections are due to the antenna itself, the so-called self-interference signal. Isolation and reflection will not degrade significantly if an object is not placed in the vicinity of the antenna. Specifically, for the 5G 3 GHz frequency, the free space path loss around 3 GHz at 1 m is 42 dB, if an object is placed at 1 m from the antenna. The two-way path loss will be 84 dB lower, which will worsen the isolation and reflection. However, it will not be that significant if the isolation is around 20–30 dB range. If a very close object is placed by the antenna, this may cause a few dB change in the isolation; however, if the objects are placed far away from the antennas, similar performance should be expected. As measured in [24], the performance of high isolation antennas in a real environment will definitely change, but the normal operation of the antenna will remain stable.

### 3.2. 2D and 3D Gains

The 3D gain of antennas was measured at frequencies of 2, 2.4, 3.4, 4.4, 5.2, and 5.8 GHz. The origin of the Cartesian coordinate system, which describes the gain of antennas, is the disk center and is shown in Figure 2, Figure 6 and Figure 11 for one-, two-, and four-port antennas, respectively. In all systems, the feeding port is on the z-axis (Port 1), and the x-axis is perpendicular to the disk. Since the radiation pattern of the antennas is similar to a dipole antenna, the antennas’ co- and cross-poles are indicated by G*_θ_* and G*_ϕ_* components. Therefore, it is expected that the antennas co-pole gain (G*_θ_*) will significantly dominate the cross-pole gain (G*_ϕ_*). Furthermore, the one- and four-port antennas are symmetric with respect to the xz-plane; thus, the patterns are measured only on the hemisphere on y > 0 space.

The simulated and measured 2D gains for the one- (Figure 19), two- (Figure 20), and four-port (Figure 21) antennas were obtained at different frequencies (2.4, 4.4, and 5.8 GHz) and on xz-, yz-, and xy-planes. As shown in Figure 19, Figure 20 and Figure 21, the measured (dashed lines) and simulated (solid lines) gains in the *θ* direction (G*_θ_* in the red curve) dominate the gains in the *ϕ* direction (G*_ϕ_* in the blue curve). The simulated gains for G*_ϕ_* are smaller than the measured values, which could be due to the antennas’ imprecise alignment in the measurement setup or the AUT (antenna under test) tilt during the measurement. When the antenna gain at one pole is much smaller than the other pole, some tilt in AUT can cause a significant increase in the value of the cross-pole. The effect of the feed cable is seen in the measured G*_θ_* gain at around *θ* = 180° and on the xz- and yz-planes. The radiation patterns show an isotropic antenna characteristic on the xy-plane. The electric field component of G*_θ_* gain on the xz-plane is perpendicular to disk at *θ* = 0° and 180° and results in null at these angles. Moreover, the fact that the antennas radiation at *θ* = 0° and 180° is lower (smaller gain) than at other angles shows the antennas behave very similar to a dipole antenna.

The 3D gain of the antenna at 4.4 GHz for total gain, *θ* polarization, and *ϕ* polarization are shown in Figure 22. As discussed, the level of G*_θ_* is higher than G*_ϕ_* for all of the antennas. For all ports of the antenna, nulls are seen around the antenna feeding ports. One can note that the antenna almost radiates in the available space, making an ideal antenna for MIMO wireless systems.

### 3.3. Gain and Efficiency versus Frequency

The gains of the antennas are measured in some specific frequencies including 2, 2.4, 3.4, 4.4, 5.2, and 5.8 GHz, as shown in Figure 23a. The agreement between the measured and simulated gains of the antennas is decreased as the number of the ports is reduced. Since the single-port antenna radiates in all directions, the absorbers near the antenna in the anechoic chamber change the antenna performance and specifically in the lower frequencies. Note that the ground plane around the two- and four-port antennas reduces the antenna radiation on the antenna plane (xz-plane), in the direction that the near absorbers to the AUT are positioned. Consequently, the destructive effect of these absorbers, near the two- and four-port antennas, are partially canceled and the measurements get closer to simulated results. 

The antennas’ measured gain varies from 3.08 dBi at 3.4 GHz for a single port to 6.74 dBi at 5.8 GHz for four-port antennas.

The radiation efficiency of the antennas is calculated using the measured (with 2° angular spacing) average 3D gain technique. The calculated efficiency plot is presented in Figure 23b. Due to the big structure of the antennas, the accuracy of the simulated radiation efficiencies is low. As the frequency is increased, each antennas’ radiation efficiency is increased. The values change from 50% to 100 %.

Table 2 presents the comparison of proposed antennas with the reported sub-6 GHz MIMO antennas in [25,26,27,28,29,30]. The presented two- and four-port antennas in this work outperform previously published sub-6 GHz antennas for 5G applications in [25,26,27,28,29,30] with the larger bandwidth and single radiating element, for which the antenna size, gain, efficiency, and performance in MIMO applications are comparable to existing studies where multiple radiating elements are used.

## 4. Conclusions

A wideband single-port antenna, with λ/4 diameter of the radiating disk (monopole-like) and dipole-like radiation pattern, was designed and manufactured. The geometry of the single-port antenna was utilized as the prototype for the two- and four-port antennas, by rotating (90°) the single-port geometry with respect to the disk center and adding a new port. The ground planes of the ports (other than the single port) were connected to improve/increase the isolation between ports. The measured and simulated data are in good agreement. The acceptable correlation coefficient during the bandwidth makes the antenna suitable for the MIMO application for the 5G NR sub-6 GHz band. Finally, the design challenge of two or four separate antennas being near each other and any potential coupling between them can be solved by these monolithic compact antennas that contain good matching, proper isolation between the ports and omnidirectional-like radiation pattern. The antennas are not only very compatible, but their reflection coefficient/isolation between the ports can be further improved, to achieve even better values (for a limited bandwidth or single operation between 2 and 6 GHz), by optimizing the dimensions of the introduced parameters (assuming prior knowledge of their influence on Sii and Sij).

## Figures and Tables

**Figure 1 sensors-20-06960-f001:**
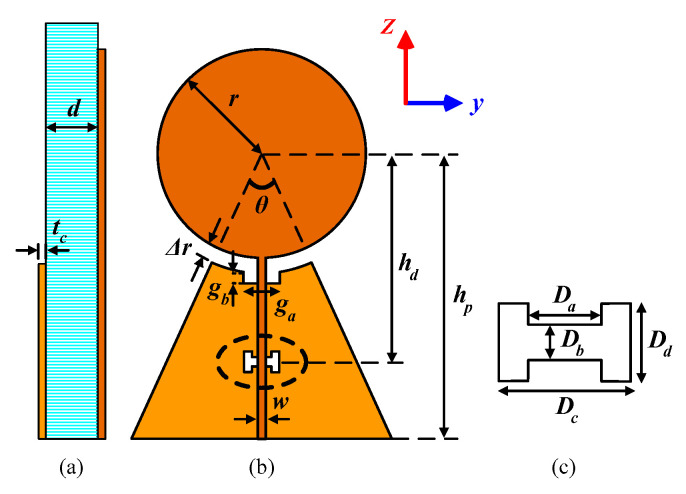
Schematic for the proposed single port wideband disk antenna: (**a**) cross-sectional view; (**b**) top view; and (**c**) etched ground dimensions.

**Figure 2 sensors-20-06960-f002:**
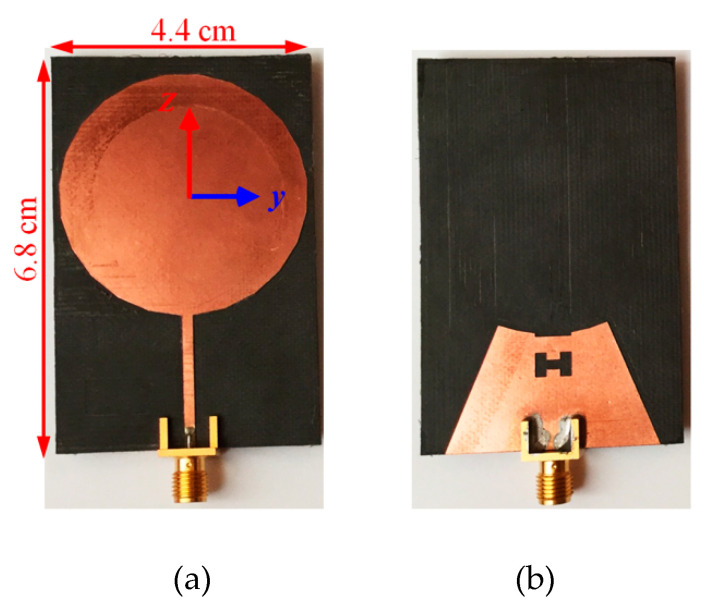
Fabricated single-port antenna: (**a**) top view; and (**b**) bottom view.

**Figure 3 sensors-20-06960-f003:**
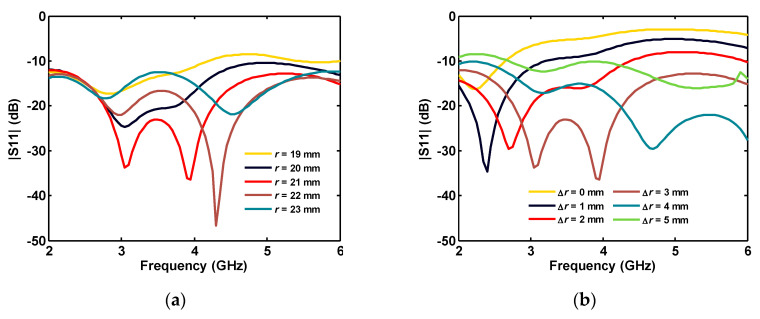
Single-port antenna simulated reflection coefficient for different: (**a**) disk radii; and (**b**) the gap length between disk and the ground plane.

**Figure 4 sensors-20-06960-f004:**
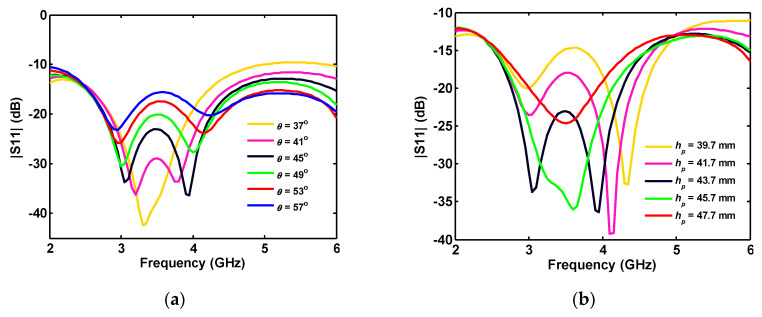
Single-port antenna simulated reflection coefficient in terms of triangle parameters: (**a**) angle; and (**b**) height.

**Figure 5 sensors-20-06960-f005:**
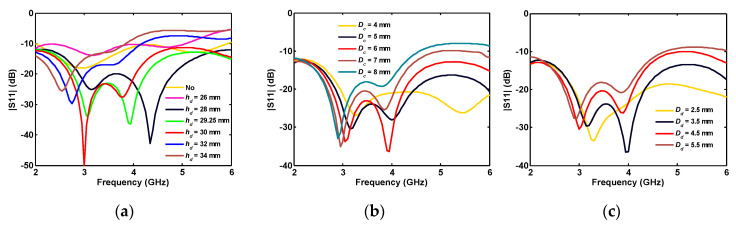
Single-port antenna simulated reflection coefficient for different (**a**) location; (**b**) width; and (**c**) height of the dumbbell-shaped etched ground.

**Figure 6 sensors-20-06960-f006:**
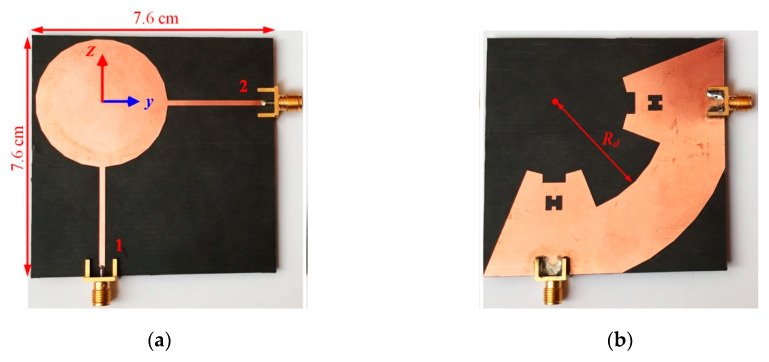
Fabricated two-port antenna: (**a**) top view; and (**b**) bottom view.

**Figure 7 sensors-20-06960-f007:**
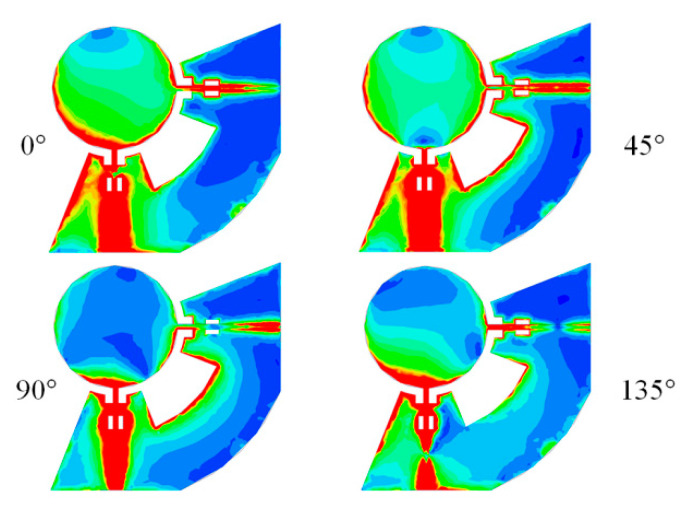
The simulated magnitude of current distribution on top and bottom layers at 2.4 GHz and different 0-, 45-, 90-, and 135-degree phases.

**Figure 8 sensors-20-06960-f008:**
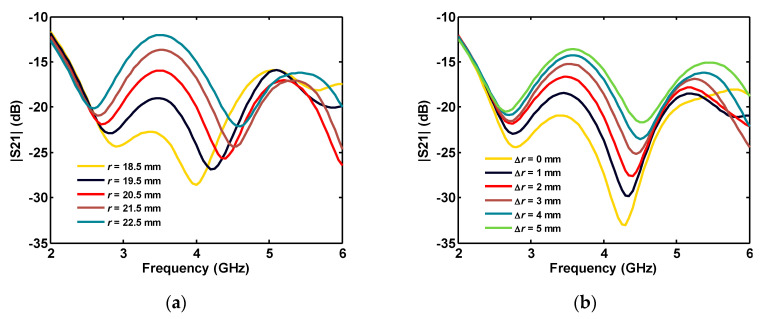
Simulated insertion loss between the ports of two-port antenna for various: (**a**) disk radius; and (**b**) disk-ground spacing.

**Figure 9 sensors-20-06960-f009:**
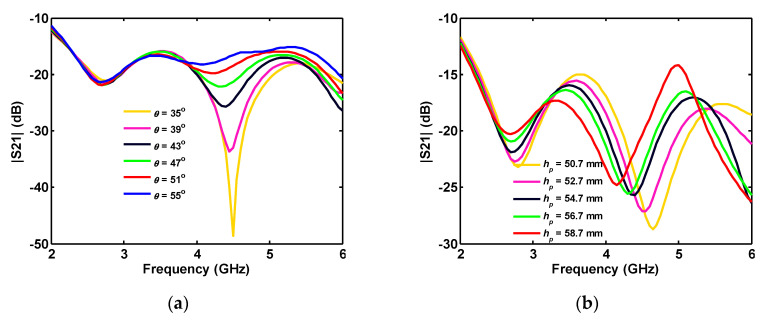
Simulated insertion loss between the ports of two-port antenna for various: (**a**) angle; and (**b**) height of the truncated triangular-shaped ground plane.

**Figure 10 sensors-20-06960-f010:**
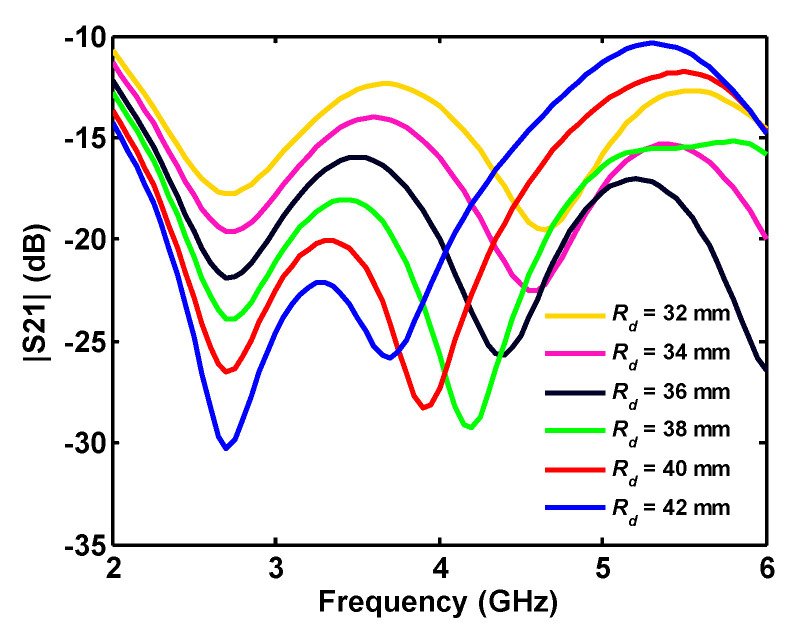
Effect of the inner radius of the ground plane sector on simulated insertion loss.

**Figure 11 sensors-20-06960-f011:**
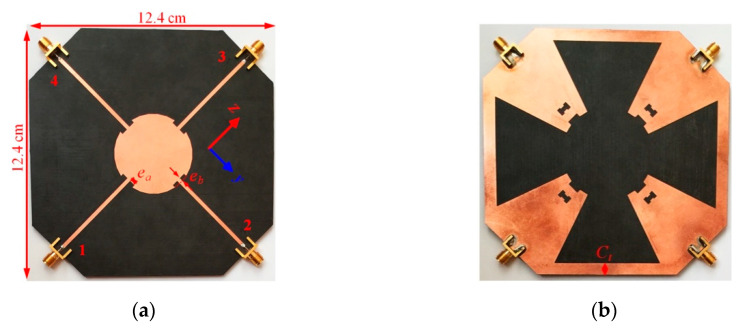
Fabricated four-port antenna: (**a**) top view; and (**b**) bottom view.

**Figure 12 sensors-20-06960-f012:**
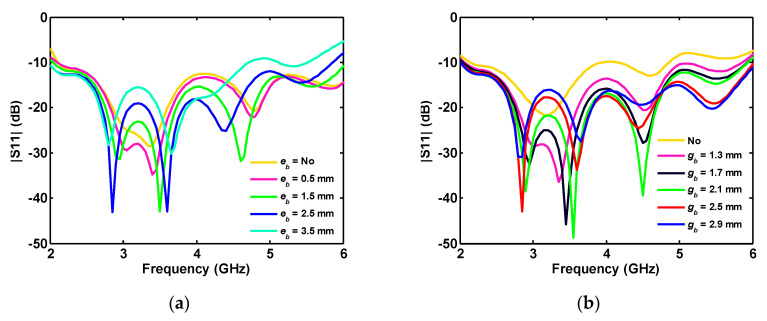
Effect of etched area’s simulated reflection coefficient of four-port antenna: (**a**) disk; and (**b**) edge of the ground plane.

**Figure 13 sensors-20-06960-f013:**
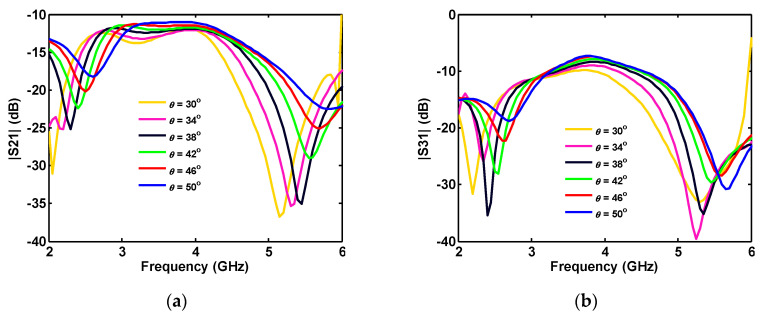
Effect of ground plane angle *θ* on the simulated isolation (a) S_21_ and (b) S_31_ between different ports of four-port antenna.

**Figure 14 sensors-20-06960-f014:**
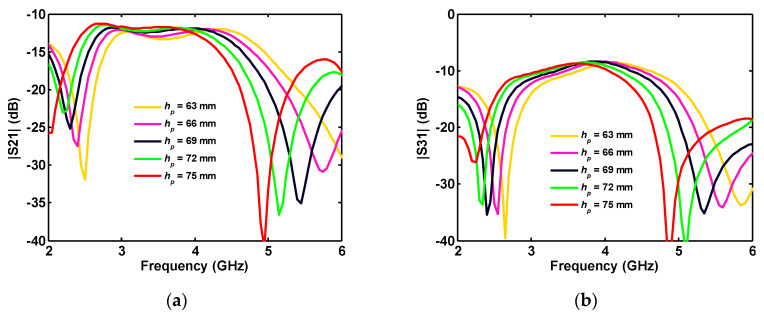
Effect of the ground planes height on the different ports’ isolation (**a**) S_21_ and (**b**) S_31_ in a four-port antenna.

**Figure 15 sensors-20-06960-f015:**
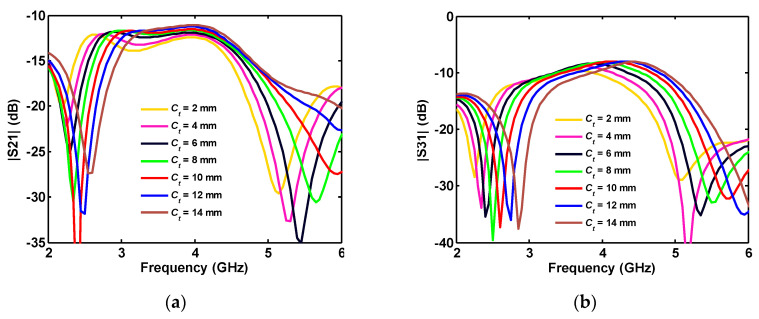
Influence of the ground joint thickness on the different ports simulated isolation (a) S_21_ and (b) S_31_ in four-port antenna.

**Figure 16 sensors-20-06960-f016:**
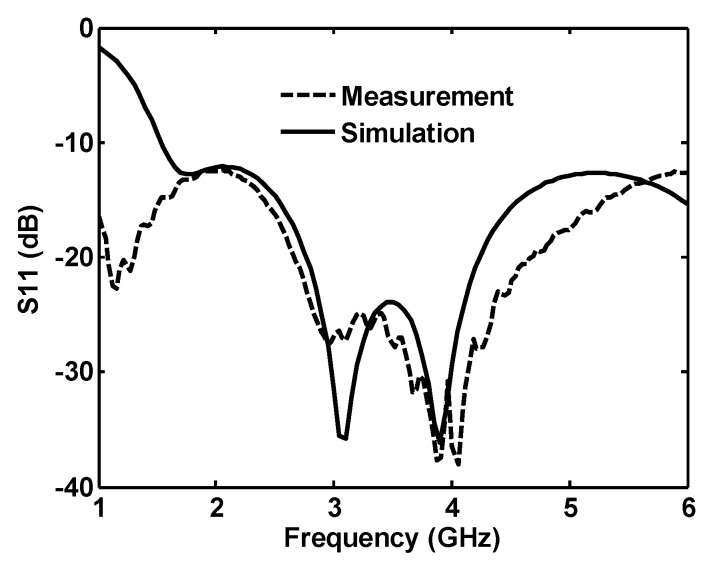
Measured and simulated reflection coefficient for the single-port antenna.

**Figure 17 sensors-20-06960-f017:**
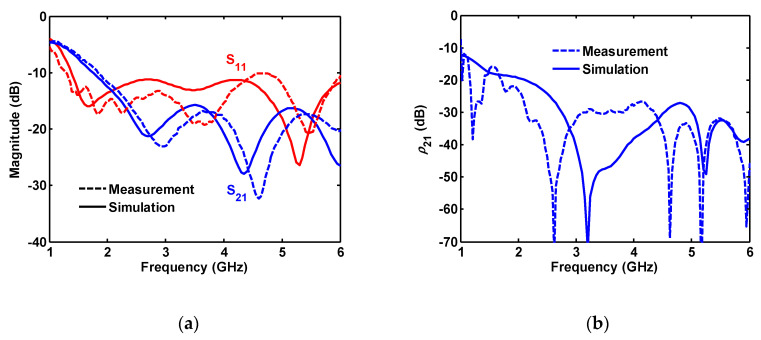
Measured and simulated (**a**) S-parameters and (**b**) correlation coefficient for two-port antenna.

**Figure 18 sensors-20-06960-f018:**
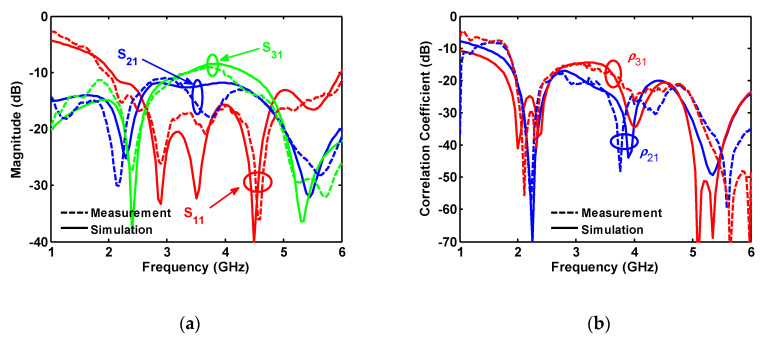
Measured and simulated (**a**) S-parameters and (**b**) correlation coefficient for four-port antenna.

**Figure 19 sensors-20-06960-f019:**
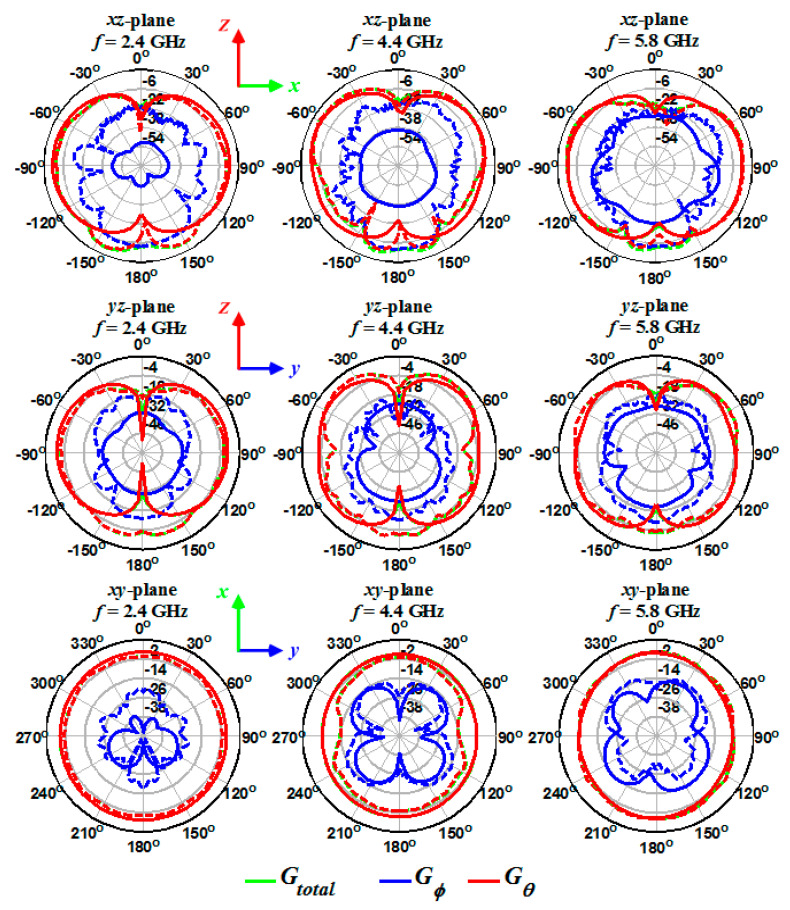
Measured (dashed lines) and simulated (solid lines) gain for the single-port antenna on three different planes and frequencies.

**Figure 20 sensors-20-06960-f020:**
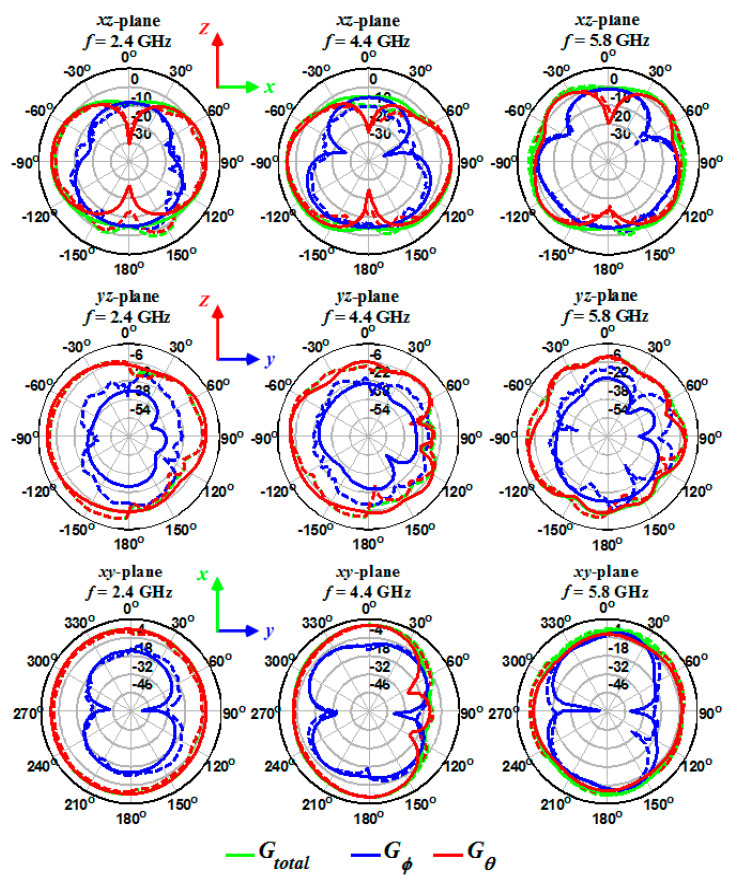
Measured (dashed lines) and simulated (solid lines) gain for the two-port antenna on three different planes and frequencies.

**Figure 21 sensors-20-06960-f021:**
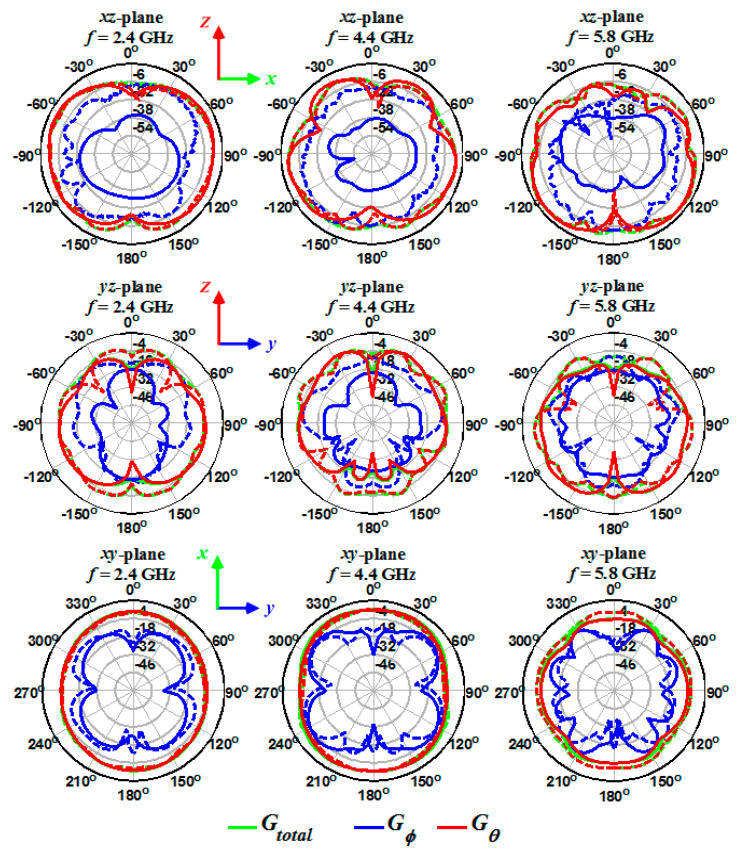
Measured (dashed lines) and simulated (solid lines) gain for the four-port antenna on three different planes and frequencies.

**Figure 22 sensors-20-06960-f022:**
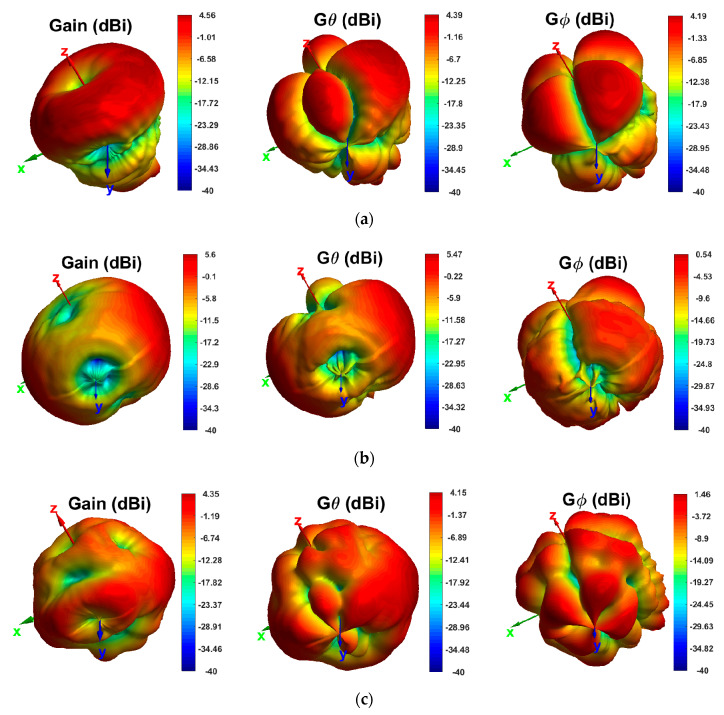
Measured 3D total, *θ*-polarized and *ϕ*-polarized gains at 4.4 GHz for: (**a**) single-port antenna; (**b**) two-port antenna; and (**c**) four-port antenna.

**Figure 23 sensors-20-06960-f023:**
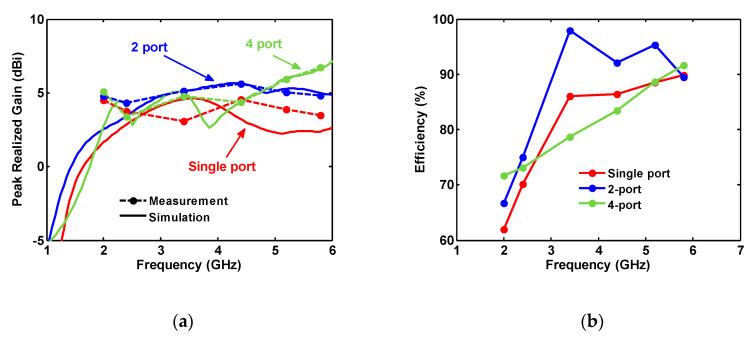
(**a**) Measured and simulated peak gain; and (**b**) measured antenna efficiency values.

**Table 1 sensors-20-06960-t001:** Geometrical dimensions for different antennas.

	Single Port	Two-Port	Four-Port
*h_p_*	43.7	54.7	69
*r*	21	20.5	20
Δ*r*	3	2.5	3
*θ*	45	43	38
*w*	1.8	1.8	2
*g_a_*	0.6	2.8	2.1
*g_b_*	6	7	6
*h_d_*	29.25	32	31
*D_a_*	2.4	1.4	2.4
*D_b_*	1.5	2	2
*D_c_*	6	5	6
*D_d_*	4	4.5	3.5
*R_d_, C_t_*	-	*R_d_* = 36	*C_t_* = 6
*e_a_*	-	-	3
*e_b_*	-	-	2

Values are in mm.

**Table 2 sensors-20-06960-t002:** Comparison of proposed antennas in this work and antennas in [25,26,27,28,29,30].

Ref.	Port (#)	Radiating Element	Size (cm)	Freq. (GHz)	BW Def.	BW (MHz)	Isolation (dB) >	ECC	Gain (dBi)
[25]	4	Inverted-F antennas	8.8 × 8.8	2.06–2.16	10 dB	100	22 Meas.	0.05	4.34
[25]	6	Inverted-F antennas	16 × 14	2.2–2.3	10 dB	100	24 Meas.	0.01	4.29
[26]	8	Fork-shaped dipoles, L-Shaped	13 × 10	3.6–3.83 5.15–5.925	10 dB	230 775	15 Sim.	0.070.03	2.6–3.1 2.5–4.2
[27]	4	F-shaped elements	15 × 7.5 × 0.4	3.3–4.2 4.4–5.0 5.15–5.85	8 dB	900 600 700	14 Meas.	0.05	4.39 3.66 4.62
[21]	8	Rectangular monopoles	13.6 × 6.8 × 0.62	3.6–4.2 4.4–4.9 5.15–5.925	6 dB	600 500 775	10 Meas.	0.1	-
[28]	4	Inverted L monopoles	4 × 4	2.7–4.94	10 dB	2240	11 Meas.	0.1	4
[29]	4	Elliptical ring slot antennas	12.7 × 7	3.4–3.8	10 dB	400	20 Sim.	0.01	1
[30]	2	Rhombus -shaped antennas	2 × 3.5	3.34–3.87	10 dB	530	15 Meas.	0.01	2.34
[30]	12	Rhombus -shaped antenna	20.6 × 11.5	3.4–3.8	10 dB	400	15 Sim.	-	3.2
This work	2	Single Disk	7.6 × 7.6	2.3–6.0	10 dB	3700	15 Meas.	0.01	5
This work	4	Single Disk	12.4 × 12.4	2.0–6.0	10 dB	4000	10 Meas.	0.03	3.1–6.75

Abbreviations: Ref., References; BW, Bandwidth; BW Def., Bandwidth Definition; ECC, Envelope Coefficient Coefficient.

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
