# Peer review of "Wideband Multiport Antennas"

_sensors, 2020, doi:10.3390/s20236960_

Round 1

Reviewer 1 Report

The paper is interesting. The topics seems to be well explained with great details and great care in illustrating, through the use of graphs, the results obtained. 

In any case, some clarifications and some corrections seems to be necessary.

Fig. 2 the colour of the symbol and text may be improved 

Fig. 3 clarify in section 2 if the results are simulations or measures 

Fig. 4 clarify in section 2 if the results are simulations or measures 

Fig. 5 clarify in section 2 if the results are simulations or measures

Fig. 6 the colour of the symbol and text may be improved 

Fig. 8 clarify in section 2 if the results are simulations or measures 

Fig. 9 clarify in section 2 if the results are simulations or measures 

Fig. 10 clarify in section 2 if the results are simulations or measures

Fig. 11 the colour of the symbol and text may be improved 

Line 209 explain better how the antenna may be scaled in dimention for mobile application.

Fig. 19 the figure have to be improved in terms of distribution. The single images are too small and difficult to understand

Fig. 20 the figure have to be improved in terms of distribution. The single images are too small and difficult to understand

Conclusions: The content has to be improved and became more detailed  especially with reference to 5G applications

Academic Editor Notes

A wideband four-port monopole antenna is proposed for sub 6 GHz MIMO applications. The paper is well-structured. The measured and simulated results are provided and discussed. A good agreement is obtained between simulation and measurements. The proposed antenna design is new with a modified ground plane. The mutual coupling of the elements is less than -10 dB over the operation band of the antenna.

All review comments should be clearly/carefully addressed.

We would like thank Reviewers for their valuable comments, we believe the paper is now better explained and compared with similar antennas for sub 6GHz band with their comments and questions.

Authors should modify the text and highlight the novelty of the proposed design.

We have modified the text and highlight the novelty of the proposed design.

Line 69:

We propose a structure to increase the number of radiating element feeding/receiving ports only by rotating the main single port monopole antenna. Of course, monopole antenna is well known and there are many reports on how to make it wideband, however, increasing the number of ports while matching the ports and decreasing coupling between the ports need a lot of attempts. Moreover, when all the ports are using common radiating elements it needs some smart methods to mitigate coupling between the ports. In this paper, we use a unit structure and bridge between the ground planes of ports to alleviate coupling between ports. We have designed and optimized the antenna for frequency band between 2 and 6 GHz and achieved min isolation of 10 dB between the four ports. The aim is to introduce a multi-purpose (multiport and wideband) structure, however, for the desired application/band the isolation between ports can be increased only by optimizing the ground plane and connection between ports.

This was already in the text to mention about the novelty of the antennas.

Line 81:

The four port antenna reported in this paper can be used for a multi-frequency system requiring many antennas. 4x4 MIMO implemented for a WLAN on 2.4 and 5.2 GHz band can be one example. The four port antenna can also be used for the sub-6GHz band 5G system. For a multiple radio system currently used in smartphones, let’s assume there are 4 radios and these radios are a 3G (2GHz band), WLAN (2.4 GHz), 1-6GHz 5G (3.6 GHz band), and WLAN (5.2 GHz). One can directly connect these 4 radios to the proposed four port antenna without any switches and duplexers. RF filters can be deployed for each radio band to provide enough selectivity. However, with our antenna, all these radios can operate simultaneously. The key focus is on new mobile 5G bands including spectrum in the 3.5 GHz range that has been assigned in numerous countries. However, several countries including China and Japan plan to use spectrum in the 4.4-4.9 GHz range for 5G in addition to a growing number of countries considering the 3.5- 4.2 GHz range and the 2.3 GHz and 2.5/2.6 GHz bands for 5G NR [21].

The authors are also recommended to add a table to compare the fundamental characteristics of the proposed antenna design with recently published papers.

We have prepared a table (Table II.) which compares the fundamental characteristics of our antennas with similar 5G sub 6 Ghz antennas from recently published papers and we see that our antenna size is comparable and our performance in terms of bandwidth is better.

Reviewer 2 Report

1) The real application scenarios should be added, as such two or four port antennas is easily affected by the surroudings.

2) The novelty of the antennas design method should be highlighted. In current version, lots of full-wave simulations results is presented, and the novelty is poor.

3) The isloation between each two ports seems not enough for MIMO application.

4) The size of the antennas seems too big.

Academic Editor Notes

A wideband four-port monopole antenna is proposed for sub 6 GHz MIMO applications. The paper is well-structured. The measured and simulated results are provided and discussed. A good agreement is obtained between simulation and measurements. The proposed antenna design is new with a modified ground plane. The mutual coupling of the elements is less than -10 dB over the operation band of the antenna.

All review comments should be clearly/carefully addressed.

We would like thank Reviewers for their valuable comments, we believe the paper is now better explained and compared with similar antennas for sub 6GHz band with their comments and questions.

Authors should modify the text and highlight the novelty of the proposed design.

We have modified the text and highlight the novelty of the proposed design.

Line 69:

We propose a structure to increase the number of radiating element feeding/receiving ports only by rotating the main single port monopole antenna. Of course, monopole antenna is well known and there are many reports on how to make it wideband, however, increasing the number of ports while matching the ports and decreasing coupling between the ports need a lot of attempts. Moreover, when all the ports are using common radiating elements it needs some smart methods to mitigate coupling between the ports. In this paper, we use a unit structure and bridge between the ground planes of ports to alleviate coupling between ports. We have designed and optimized the antenna for frequency band between 2 and 6 GHz and achieved min isolation of 10 dB between the four ports. The aim is to introduce a multi-purpose (multiport and wideband) structure, however, for the desired application/band the isolation between ports can be increased only by optimizing the ground plane and connection between ports.

This was already in the text to mention about the novelty of the antennas.

Line 81:

The four port antenna reported in this paper can be used for a multi-frequency system requiring many antennas. 4x4 MIMO implemented for a WLAN on 2.4 and 5.2 GHz band can be one example. The four port antenna can also be used for the sub-6GHz band 5G system. For a multiple radio system currently used in smartphones, let’s assume there are 4 radios and these radios are a 3G (2GHz band), WLAN (2.4 GHz), 1-6GHz 5G (3.6 GHz band), and WLAN (5.2 GHz). One can directly connect these 4 radios to the proposed four port antenna without any switches and duplexers. RF filters can be deployed for each radio band to provide enough selectivity. However, with our antenna, all these radios can operate simultaneously. The key focus is on new mobile 5G bands including spectrum in the 3.5 GHz range that has been assigned in numerous countries. However, several countries including China and Japan plan to use spectrum in the 4.4-4.9 GHz range for 5G in addition to a growing number of countries considering the 3.5- 4.2 GHz range and the 2.3 GHz and 2.5/2.6 GHz bands for 5G NR [21].

The authors are also recommended to add a table to compare the fundamental characteristics of the proposed antenna design with recently published papers.

We have prepared a table (Table II.) which compares the fundamental characteristics of our antennas with similar 5G sub 6 Ghz antennas from recently published papers and we see that our antenna size is comparable and our performance in terms of bandwidth is better.

Round 2

Reviewer 2 Report

1) The "21" in "S21" should be with the subscript format.  

2) Different curves in one figure should with different line type or different graphics dot, not only with different color.

Author Response

1) The "21" in "S21" should be with the subscript format.  

These are updated in the text. Thanks.

2) Different curves in one figure should with different line type or different graphics dot, not only with different color.

To show the antenna performance for changes in different geometrical dimensions of antenna, many number of curves are plotted in a single figure. We have used color marks while there are limited line marks (keeping the lines thickness and avoiding figure complexity) to correspond to each of the curves in a plot (belonging to simulation results). 

Nevertheless, after Figure 16 which we present and compare measurement and simulation results we use both color and line marks to track simulation and corresponding measurement easily.